# Dynamic modulation of pulsatile activities of oxytocin neurons in lactating wild-type mice

Kasane Yaguchi[1,2], Mitsue Hagihara[1], Ayumu Konno[3,4], Hirokazu Hirai[3,4], Hiroko Yukinaga[1], Kazunari Miyamichi[1]*

1 Laboratory for Comparative Connectomics, Riken Center for Biosystems Dynamics Research, Kobe, Hyogo, Japan, 2 Graduate School of Biostudies, Kyoto University, Kyoto, Kyoto, Japan, 3 Gunma University Graduate School of Medicine, Maebashi, Gunma, Japan, 4 Viral Vector Core, Gunma University Initiative for Advanced Research (GIAR), Maebashi, Gunma, Japan

* kazunari.miyamichi@riken.jp

**Data Availability Statement:** All relevant data are within the paper.

## Abstract

Breastfeeding, which is essential for the survival of mammalian infants, is critically mediated by pulsatile secretion of the pituitary hormone oxytocin from the central oxytocin neurons located in the paraventricular and supraoptic hypothalamic nuclei of mothers. Despite its importance, the molecular and neural circuit mechanisms of the milk ejection reflex remain poorly understood, in part because a mouse model to study lactation was only recently established. In our previous study, we successfully introduced fiber photometry-based chronic imaging of the pulsatile activities of oxytocin neurons during lactation. However, the necessity of Cre recombinase-based double knock-in mice substantially compromised the use of various Cre-dependent neuroscience toolkits. To overcome this obstacle, we developed a simple Cre-free method for monitoring oxytocin neurons by an adeno-associated virus vector driving GCaMP6s under a 2.6 kb mouse *oxytocin* mini-promoter. Using this method, we monitored calcium ion transients of oxytocin neurons in the paraventricular nucleus in wild-type C57BL/6N and ICR mothers without genetic crossing. By combining this method with video recordings of mothers and pups, we found that the pulsatile activities of oxytocin neurons require physical mother–pup contact for the milk ejection reflex. Notably, the frequencies of photometric signals were dynamically modulated by mother–pup reunions after isolation and during natural weaning stages. Collectively, the present study illuminates the temporal dynamics of pulsatile activities of oxytocin neurons in wild-type mice and provides a tool to characterize maternal oxytocin functions.

## Introduction

Oxytocin (OT) is a nine-amino-acid peptide hormone known to mediate uterine contractions during parturition and milk ejection during lactation [1–3]. This hormone is predominantly produced by the OT neurons located in the paraventricular (PVH) and supraoptic (SO) hypothalamus and secreted into the circulation via the posterior pituitary. The milk ejection reflex, that is, the active transfer of milk from alveolar storage to mammary ducts in response to a

**Funding:** This study was supported by the program for Brain Mapping by Integrated Neurotechnologies for Disease Studies (Brain/MINDS, JP21dm027111) from Japan Agency for Medical Research and Development (AMED, https://brainminds.jp/en/) to H.H. and by KAKENHI (20K20589 and 21H02587) from the Japan Society for the Promotion of Science (JSPS, https://www.jsps.go.jp/english/e-grants/index.html) to K.M. The funders had no role in study design, data collection and analysis, decision to publish, or preparation of the manuscript.

**Competing interests:** The authors have declared that no competing interests exist.

transient increase in plasma OT, is disabled in *OT* or *OT receptor* (*OTR*) knockout postpartum mice [4–7]. Therefore, OT is indispensable for breastfeeding in mice. The secretion of OT is thought to be mediated by the synchronous burst activities of PVH and SO OT neurons [5]. Extracellular recording studies have characterized maternal OT neural activities [8–12] and described afferent circuitry that conveys nipple sensory stimuli to the OT neurons in the hypothalamus [13–15]. Intracerebroventricular injection of OT is known to facilitate the milk ejection reflex, whereas that of OTR antagonist can block the ongoing milk ejection [16, 17]. Despite these classical studies, which were conducted in rats and rabbits, the detailed molecular and neural circuit mechanisms by which OT neurons shape burst synchronous activities during the milk ejection reflex remain poorly understood, in part because it has been difficult to utilize cell-type-specific toolkits for the manipulation of gene functions and neural activities in these species.

Our recent study demonstrated cell type-specific calcium ion ($Ca^{2+}$) imaging of PVH OT neurons in parturient and lactating mother mice by fiber photometry [18]. In that study, we utilized *OT-Cre* mice [19] combined with the Cre-dependent GCaMP6s driver line, *Ai162* [20]. Although these double knock-in mice allowed cell type-specific and intensive expression of GCaMP6s, this strategy compromises the use of various Cre-dependent neuroscience toolkits. Applying a simple Cre-free method for monitoring OT neural activities [21] to mice could overcome this problem. In addition, previous studies [18, 21] have analyzed only the early stages of lactation; therefore, the dynamics of the pulsatile activities of OT neurons throughout the different stages of lactation remain unknown.

In the present study, we first aimed to develop an adeno-associated virus (AAV) vector to drive GCaMP6s selectively into the OT neurons via a mouse *OT* mini-promoter [22]. Then, after validating this system, we aimed to characterize the pulsatile activities of OT neurons quantitatively in different mouse strains and multiple lactation stages, including weaning stages. In addition, we aimed to investigate whether direct physical contact between mothers and pups is always required for the milk ejection reflex in mice. As plasma OT levels are increased even before physical suckling in well-experienced lactating women [23] and livestock animals such as ewes [24], we analyzed the activity patterns of OT neurons in mother mice in the middle lactation stage upon mother–pup physical separation and reunion. The results indicated that the pulsatile activities of OT neurons in mice require physical mother–pup contact and that the frequency of pulsatile activities can be dynamically modulated by multiple factors.

## Materials and methods

### Animals

Animals were housed under a 12-h light/12-h dark cycle with food and water supplied ad libitum. Wild-type C57BL/6N and ICR mice were purchased from JAPAN SLC (Hamamatsu, Japan). All experimental procedures were approved by the Institutional Animal Care and Use Committee of the RIKEN Kobe branch (A2017-15-14).

### Viral preparations

To construct *pAAV-OTp-GCaMP6s*, we performed PCR using the C57BL/6J mouse genome as a template with the PCR primers 5'-agatgagctggtgagcatgtgaagacatgc and 5'-ggcgatggtgctcagatccgctgt to subclone a 2.6-kb mouse *OT* promoter (*OTp*) [22], which is orthologous to rat *OTp* [25]. *pAAV.CAG.Flex.GCaMP6s.WPRE.SV40* (Addgene #100842) was used as a PCR template to subclone the *GCaMP6s* cassette. The AAV serotype 9 *OTp-GCaMP6s* was created at the Gunma University Viral Vector Core using the

ultracentrifugation method, as described previously [26]. *pAAV-OTp-GCaMP6s* is available in Addgene (#192945).

## Stereotactic injection

For targeting AAV into a certain brain region, stereotactic coordinates were first defined based on the mouse brain atlas [27]. Mice were anesthetized with 65 mg/kg ketamine (Daiichi-Sankyo) and 13 mg/kg xylazine (Sigma-Aldrich) via intraperitoneal injection and head-fixed to the stereotactic equipment (Narishige). For fiber photometry, 200 nl of AAV9 *OTp-GCaMP6s* (titer: $2.9 \times 10^{12}$ genome particles per ml) was injected into the PVH at the following coordinates: 0.6 mm anterior from the bregma, 0.3 mm lateral from the bregma, and 4.5 mm ventral from the brain surface. After the viral injection, the incision was sutured, and the animal was allowed to recover from anesthesia. The animal was then returned to the home cage.

## Fiber photometry

Fiber photometry recordings were conducted as described previously [18]. Briefly, a 400-μm core, 0.5 NA optical fiber (Thorlabs, cat#FP400URT) was implanted immediately above the PVH of the AAV-injected wild-type adult female mice (age 2–5 months). After the surgery, the female mouse was crossed with a male of the same strain and housed in the home cage until recording. The neurons expressing GCaMP were illuminated by 465-nm (modulated at 309.944 Hz) and 405-nm (modulated at 208.616 Hz) lights. Emitted fluorescence was collected using the integrated Fluorescence Mini Cube (Doric Lenses, iFMC4_AE (405)_E (460–490)_F (500–550)_S). Light collection, filtering, and demodulation were performed using Doric photometry setup and Doric Neuroscience Studio Software (Doric Lenses). The 405-nm signal was recorded as a background (non–calcium-dependent), and the 465-nm signal was used as a calcium-dependent GCaMP6s response. The power output at the tip of the fiber was approximately 5 μW by a photodiode power sensor (S120VC, Thorlabs). The signals were initially acquired at 12 kHz and then decimated to 120 Hz. We used a 5-Hz low-pass filter before the analysis of the photometric peaks of OT neurons.

For the analyses, we used custom-made Python code. Briefly, the background 405-nm signals were subtracted from the 465-nm signals after being fitted by the least-squares method. The ΔF/F (%) was then calculated as $100 \times (Ft–F0) / F0$, where F0 was the average of the background-subtracted signals over the whole recording period, and F$t$ was the background-subtracted signal at time = $t$. The height of the peaks in each mother varied considerably partly because the optical fiber location relative to the PVH was variable; therefore, to identify the photometric peaks reliably, we first selected several visually obvious peaks to estimate the peak height of that animal. The photometric peaks of the OT neurons were then automatically detected by using the findpeaks function in the Scipy module in Python, with the peak threshold of half the estimated peak height, and the full-width at half-maximum (FWHM) threshold over 1 s. To show the peri-event traces of the peaks, we first defined the peak as the local maximum point of ΔF/F, and time 0 as the point when the ΔF/F value reached half the peak height. We then extracted the ΔF/F data from –10 s to +15 s around the time 0 and adjusted the median fluorescence of the –8 to –3 s baseline period to zero to align multiple data points along the y-axis. For the analysis of photometric peaks during lactation, postpartum day (PPD) 1 was defined as 1 day after the day of parturition. To eliminate a potential effect of the number of pups on the activity patterns of OT neurons, the litter size was adjusted to n = 6 at PPD 0. If the number of pups was less than 6 or pups died during lactation, foster pups of the same age were added.

In Fig 2, PPD2–4 C57BL/6N and ICR mothers were subjected to photometry recordings for 6 h each in the light and dark phases. In Fig 3, PPD12–14 mothers were subjected to the photometry recordings. To habituate the mothers to the experimental environment, we placed two hemispherical wire mesh enclosures (5 cm in diameter) in the home cage for 2 days before the recordings. On the day of the experiment, we first conducted photometry for 2 h during the light phase to obtain the "before" data of the pulsatile activities of OT neurons. We then isolated the pups into the sealed wire mesh at the onset of the dark phase and let the mother freely investigate the wire mesh to obtain visual, auditory, and olfactory inputs from pups during physical isolation. At 6 h after the onset of the light phase on the next day, we released the pups from the wire mesh and continuously recorded photometry signals for an additional 2 h to obtain the "after" data. The entire processes were video-recorded, and the timing of the mother's investigation of the wire mesh was manually annotated. No pups died or were weakened during or after isolation, and all mothers showed breastfeeding and maternal care after the isolation. Of note, 2 out of 7 mothers in this experiment were primiparous females while 5 mothers were multiparous females (who nursed their second or third litters after weaning the previous ones).

In Figs 4B, 4C, 4E–4G, 5 and 7, we recorded photometry signals for 6 h each in the light and dark phases. For photometry recording during pup retrieval behaviors (Fig 4D), we sequentially introduced a pup to a home cage of lactating mothers at PPD12–14. We conducted photometry recording for 30 minutes per mother and observed 41 retrieval behaviors across 3 mothers. In Fig 6, we introduced foster pups of different ages to lactating mothers. First, elder pups at postnatal day (PND)17–19 were introduced to the mothers during the early lactation stage (PPD2–4). Second, younger pups at PND2–4 were presented to the mothers during the late lactation stage (PPD17–19). We monitored photometry traces for 6 hours during the light phase over three consecutive days: Day 1 with the original pups and Days 2 and 3 with the foster pups. For the fostering sessions (Days 2 and 3), we introduced the pups to the mothers in the evening of Day 1, allowing the mothers to habituate to the fostering. The original pups were kept healthy with other foster mothers and returned to their original mothers after Day 3. In Fig 7, to compare peak heights, we normalized all ΔF/Fs by the mean of the peaks at PPD18 within the same animal to obtain the normalized ΔF/F, because the exact value of ΔF/F could not be compared among different animals. PPD18 data of Fig 7 are the same as those shown in Fig 4.

Videos from an infrared camera (DMK33UX273; The Imaging Source) placed above the mouse cage were synchronized with the fiber photometry recordings. To analyze the time the mother spent in the nest, we defined the nest as follows. In most cases, the nest was visually apparent with the nest materials raised, and all pups were located within the nest materials. In rare cases in which the nest materials were not well clustered, the place where the pups were collected and gathered was defined as the nest. Based on the video recordings, we manually measured the total duration the mothers spent in the nest while their entire body was located within the nest area and in contact with the pups.

## Quantification and statistical analysis

The statistical details of each experiment, including the statistical tests used and the exact values of n, are shown in each figure legend. The p values are shown in each figure panel or legend.

To test whether the inter-peak intervals (IPIs) on PPD18 data within the cluster (Fig 5E) follow an exponential distribution, we used the least squares method to set the parameter of exponential distribution to better fit the observed IPI distributions. Then, the Kolmogorov–

Smirnov test was used to compare the experimental data with those generated by computer simulation based on the best fit exponential distribution with the data point size being equal to the experimental data size.

## Results

### An AAV vector for *OT* promoter-driving GCaMP6s

To target GCaMP6s into the OT neurons without the Cre/loxP system, we constructed an AAV vector that drives GCaMP6s under a 2.6 kb mouse *OT* promotor [6] (Fig 1A). To test the specificity of GCaMP6s expression on OT neurons using this AAV, at 2 weeks after the viral injection, coronal brain sections including the PVH were stained by *in situ* hybridization with

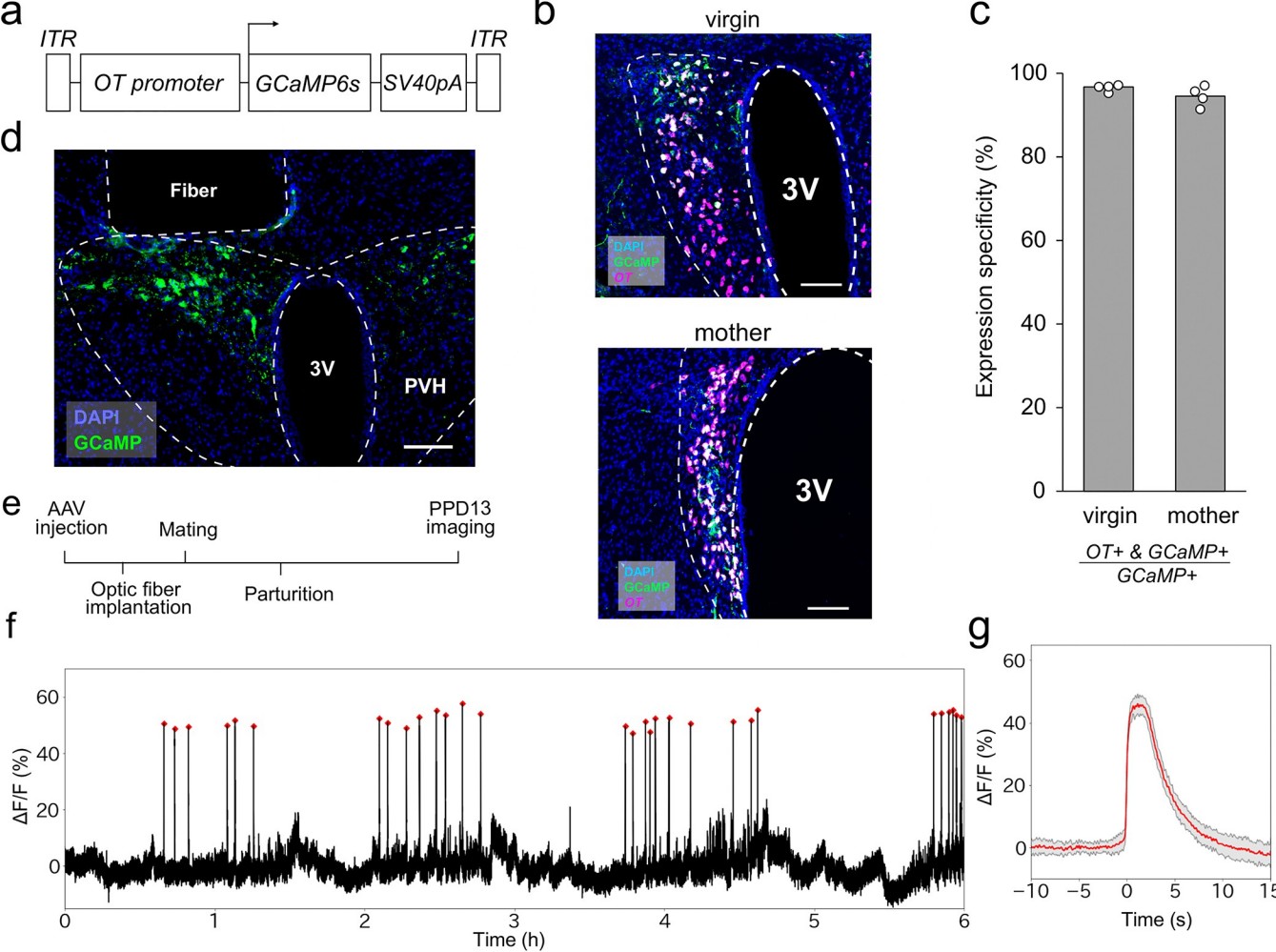

**Fig 1. Expression of AAV9 *OTp-GCaMP6s* and recording of OT neurons.** a. Schematic diagram of the AAV9 *OTp-GCaMP6s* construct. b. Typical example of a 30-μm coronal section of the PVH of virgin (top) and mother (bottom) female mice injected with *OTp-GCaMP6s* co-stained by *in situ* hybridization to detect *OT* mRNA (magenta) and anti-GFP antibody to detect GCaMP6s expression (green). Blue is nuclear staining with DAPI. c. Quantification of expression specificity (*OT*+&GCaMP6s+/GCaMP6s+) for AAV9 *OTp-GCaMP6s* in PVH OT neurons of virgin and mother female mice. White dots represent individual data (n = 4 mice), with the gray bar representing the average value. d. Representative coronal section showing the location of the optical fiber and expression of GCaMP6s (green) in the PVH. Blue is nuclear staining with DAPI. e. Schematic of the timeline of the experiment. f. Representative photometry trace showing the activities of OT neurons from a PPD13 mother for 6 h in the light phase. Red dots show the pulsatile activity of OT neurons. g. Averaged peri-event trace of the pulsatile activities (red line) observed in the 6-h recording shown in panel f, with the shadow representing the standard deviation. PVH, paraventricular hypothalamus. 3V, third ventricle. Scale bars, 100 μm.

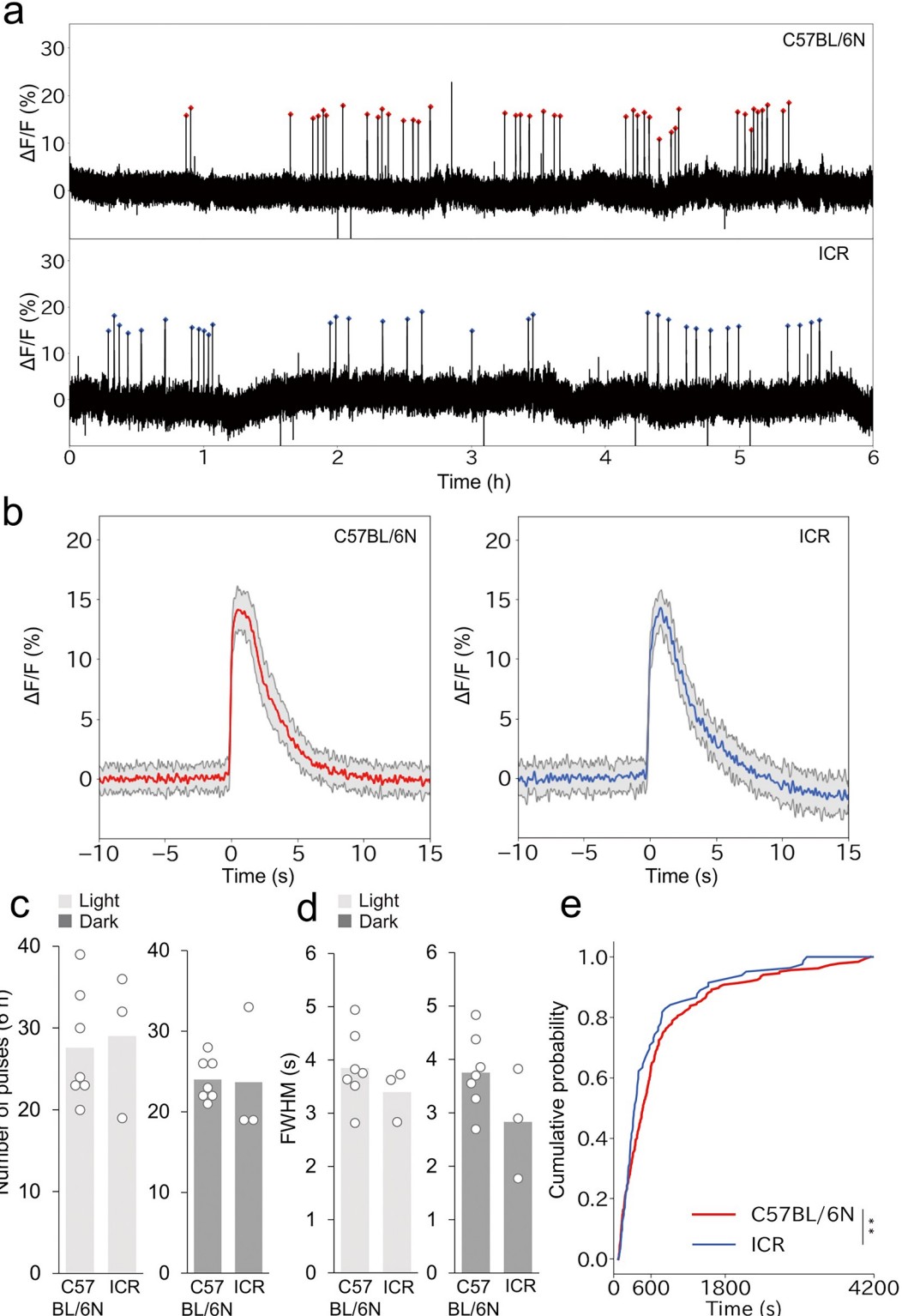

**Fig 2. Comparison of photometry traces between C57BL/6N and ICR mother mice.** a. Representative 6-h photometry traces in the light phase from PPD2–4 C57BL/6N (top) and ICR (bottom) mother mice. Red dots show the pulsatile activities of OT neurons. b. Averaged peri-event trace of photometric peaks from C57BL/6N (left, red line) and ICR (right, blue line) mothers observed in the 6-h recording shown in panel a, with shadows representing the standard deviation. c. Number of pulsatile activities of OT neurons per 6 h in C57BL/6N and ICR mother mice in the light (left) and dark (right) phases.

C57BL/6N, n = 7, ICR, n = 3. White dots represent individual data. No significance was found (p > 0.05) by a two-sided paired *t*-test (light-dark) and a two-sided Welch's *t*-test (C57BL/6N-ICR). d. Quantification of waveforms by FWHM in C57BL/6N and ICR mother mice in the light (left) and dark (right) phases. No significance was found (p > 0.05) by a two-sided paired *t*-test (light-dark) and a two-sided Welch's *t*-test (C57BL/6N-ICR). e. Cumulative distribution of IPIs of photometric peaks in C57BL/6N and ICR mother mice. **, p<0.01 by the Kolmogorov–Smirnov test.

an RNA probe for *OT*, followed by immunohistochemical staining of GCaMP6s by anti-GFP antibodies (Fig 1B). This analysis revealed that 96.7% ± 0.37% in virgin mice and 94.6% ± 1.2% in mother mice (mean ± standard error of the mean) of GCaMP6s+ neurons were *OT*+ (Fig 1B and 1C), supporting high specificity comparable to or better than the Cre-based strategy [7]. To test whether this expression of GCaMP6s is sufficient to monitor the pulsatile activities of OT neurons selectively in lactating wild-type mice by fiber photometry, we next implanted an optic fiber just above the PVH of GCaMP6s-targeted female mice (Fig 1D). After recovery from surgery, these female mice were crossed with stud male mice, and Ca$^{2+}$ imaging was performed on PPD13 (Fig 1E). The continuous 6-h photometry trace shown in Fig 1F contained 30 clear photometric peaks that highly resembled those observed in previous Cre-based strategies [7, 8]. A high-magnification view of peri-event traces (Fig 1G) showed a representative waveform of pulsatile neuronal activities of OT neurons. These data demonstrated that AAV9 *OTp-GCaMP6s* allows the recording of maternal pulsatile activities of OT neurons during lactation in mice without germline modification.

One advantage of the AAV *OTp-GCaMP6s-* compared with the Cre-based imaging strategy is that it can be easily applied to mice of any genetic background or strain. To demonstrate this utility, we performed photometry-based side-by-side recordings of PVH OT neurons for 6 h each in light and dark phases in C57BL/6N and ICR strains on PPD2–4 (Fig 2). Overall, the photometry traces were similar between the two mouse strains (Fig 2A and 2B). Next, we quantitatively analyzed the number of peaks, the cumulative distribution of the IPIs, and the waveforms of pulsatile activities of OT neurons. The number of peaks did not differ between C57BL/6N and ICR (Fig 2C). We also compared the FWHM of pulsatile activities after normalizing the peak height and found no significant differences between C57BL/6N and ICR mice (Fig 2D). By contrast, we found a slight but significant difference in the cumulative distribution of IPIs, with the ICR mothers showing higher fractions of shorter IPIs, which suggested that the temporal structures of the pulsatile activities differed between the two strains (Fig 2E). Taken together, these data confirm that *OTp-GCaMP6s*-based imaging can be applied to the ICR strain without genetic crossing, which could facilitate future studies on the pulsatile activities of OT neurons in various types of genetically modified mice.

## Characterization of pulsatile activities of OT neurons upon mother–pup physical isolation

The milk ejection reflex is initiated when sucking stimuli to the nipple of mothers are transmitted to activate the central OT neurons via the afferent neural circuitry [5]. Indeed, our previous study showed that the photometric peaks of OT neurons were not observed in the absence of pups; simultaneous suckling by three pups was required for the initiation of the pulsatile activities of OT neurons [7]. However, given that plasma OT levels are increased without physical suckling in experienced lactating women [9] and livestock animals [10], it is interesting to ask whether experienced mother mice can exhibit pulsatile activities of OT neurons when pups cannot make direct contact with their nipples. To test this, experienced C57BL/6N mother mice were first habituated with a small wire mesh enclosure in the cage for 2 days before the experiment. On the day of the experiment on PPD12–14, at 2 h after the photometry

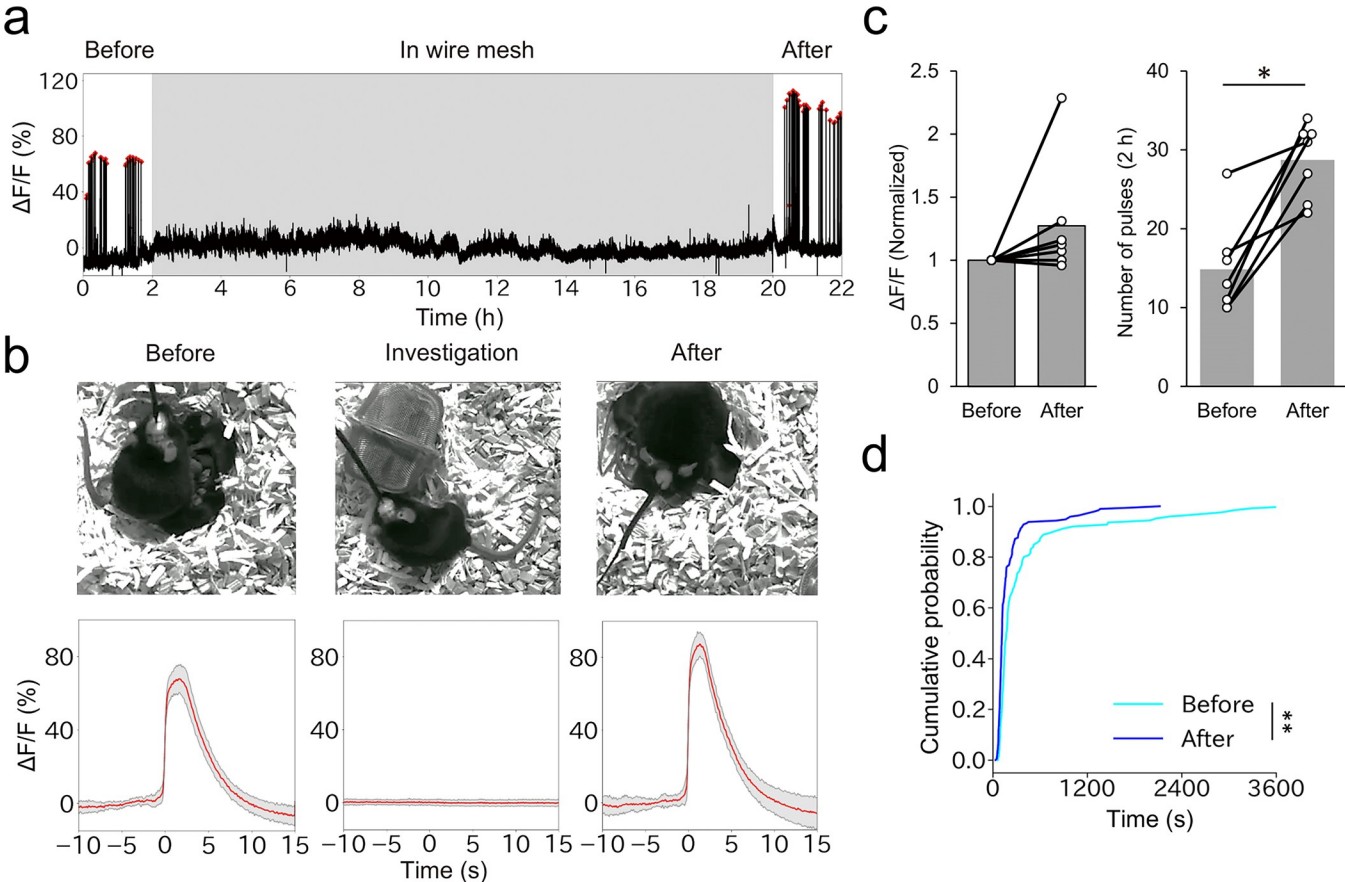

**Fig 3. Effects of mother–pup physical isolation on the activities of OT neurons.** a. Typical example of a photometry trace of PVH OT neurons in PPD12–14 mother mice before, during, and after the isolation of pups in a wire mesh. The gray shadow represents the time the pups were in the wire mesh. Red dots represent the pulsatile activities of OT neurons. b. Images of mother mice and pups during the experiment (top) and averaged peri-event traces of the photometric signals (bottom, red lines) before, during, and after the isolation of pups observed in panel a, with shadows representing the standard deviation. Time 0 in the "before" and "after" data is defined as the time point when the ΔF/F value reached half the peak, and time 0 in the "investigation" data (based on n = 202 investigation episodes) as the time point the mothers initiated an investigation of the wire mesh during isolation. c. The peak intensities defined by the normalized local maximum of ΔF/F (left) and the number of pulsatile activities of OT neurons per 2 h (right) before and after physical isolation of pups in wire mesh. n = 7 mice. *, p < 0.05 by Wilcoxon signed-rank sum test. d. Cumulative distribution of IPIs of photometric peaks before and after physical isolation of pups in wire mesh. n = 7 mice. **, p<0.01 by the Kolmogorov–Smirnov test. Of note, 2 out of 7 mothers were primiparous females while 5 mothers were multiparous females (who nursed their second or third litters after weaning the previous ones).

recording, the pups were placed into the wire mesh enclosure to prevent them from suckling. During this physical isolation, the mothers were able to investigate the wire mesh enclosure freely to obtain visual, auditory, and olfactory inputs from the pups. After isolation, the pups were returned to their normal housing condition (Fig 3A and 3B). We found no pulsatile activities of OT neurons while the pups were isolated in the wire mesh (Fig 3B), not even while they were being actively investigated by the mothers. These data suggest that direct contact by pups to the nipples is required for pulsatile activities of OT neurons, even in well-experienced mother mice.

We also found that the pulsatile activities of OT neurons were restored soon after the removal of physical isolation (Fig 3A and 3B). Interestingly, the frequency of the photometric peaks increased after recovery of lactation (Fig 3B and 3C), while the intensities remained unchanged. The number of peaks per 2-h window increased significantly, by an average of 1.9-fold (Fig 3C). This observation was supported by the cumulative distribution of IPIs,

which showed an increase of short IPIs following the isolation (Fig 3D). These observations suggest that the activities of OT neurons can be dynamically modulated depending on the preceding lactation patterns.

## Correlation between mother–pup interactions and the pulsatile activities of OT neurons

Next, we aimed to analyze the temporal dynamics of pulsatile activities of OT neurons of C57BL/6N mice during the natural lactation process, with respect to mother–pup interactions in the home cage. To this end, we video-recorded the locomotion of mothers combined with the fiber photometry recordings of PVH OT neurons and examined the correlation between the time the mother spent in the nest and the occurrence of pulsatile activities of OT neurons (Fig 4A). The animals were recorded for 6 h each in the light and dark phases in the early (PPD2–4), middle (PPD12–14), and late (PPD18) lactation stages (Fig 4B). We found that the pulsatile activities of OT neurons were closely correlated with the timing of the mothers in the nest, mostly while crouching over the pups, regardless of the light/dark phase or lactation stage (Fig 4B). The magnitude of pulsatile activity of OT neurons remained unchanged throughout the monitered lactation period in both light and dark phases (Fig 4C). In addition, no pulsatile activity of PVH OT neurons occurred during the process of the mother retrieving the pups (Fig 4D). These observations further support the hypothesis that the milk ejection reflex strictly requires direct nipple stimulation by pups. Of note, the previous study showed that the peak height increased from PPD1 to PPD12–14 [18], while no major change was observed between PPD2–4 and PPD12–14 in Fig 4C, suggesting a substantial increase in peak height during the first one or two days of lactation.

The video data revealed that the time the mothers spent in the nest during the dark phase decreased as lactation progressed (Fig 4E). Although the total number of pulsatile activities of OT neurons per 6 h of recording was constant across lactation stages (Fig 4F), the number of pulses in the dark phases normalized to the duration of the mothers staying in their nest significantly increased in the later stages of lactation (Fig 4G). By contrast, during the light phases among different lactation stages, no difference was found in the duration of the nest stay or the relative pulse frequency in the nest. These data suggest a distinct temporal dynamic of mother–pup interactions and the pulsatile activities of OT neurons between light and dark phases.

To examine quantitatively the dynamics of the pulsatile activities of OT neurons in different lactation stages, we analyzed histograms and cumulative distributions of IPIs (Fig 5A and 5B). In both the light and dark phases, the distribution significantly changed, with steeper curves of cumulative probability plots seen in the later lactation stages, suggesting that the pulse frequencies increased as lactation progressed. When the distribution of IPIs was compared between the light and dark phases, a significant difference was found in the middle and late lactation stages (Fig 5C). In these stages, shorter IPIs were found more frequently in the dark phase, suggesting a higher frequency of pulsatile activities. Because IPIs were more clearly clustered in the later lactation stages, we next aimed to characterize IPI distributions within the cluster. We defined the cluster as IPIs within 450 s because 85% of the IPIs were located within this range at PPD18. The quantile-quantile plot based on the normal distribution did not support the normality in the distributions of photometric peaks on PPD18 within the cluster (Fig 5D). We then tested whether the IPI distributions followed an exponential distribution, assuming a Poisson process. We found that the cumulative distributions of IPIs within the cluster in both the light and dark phases closely resembled those generated by computer simulations based on an exponential distribution fitted by the least-squares method (Fig 5E). These findings suggest

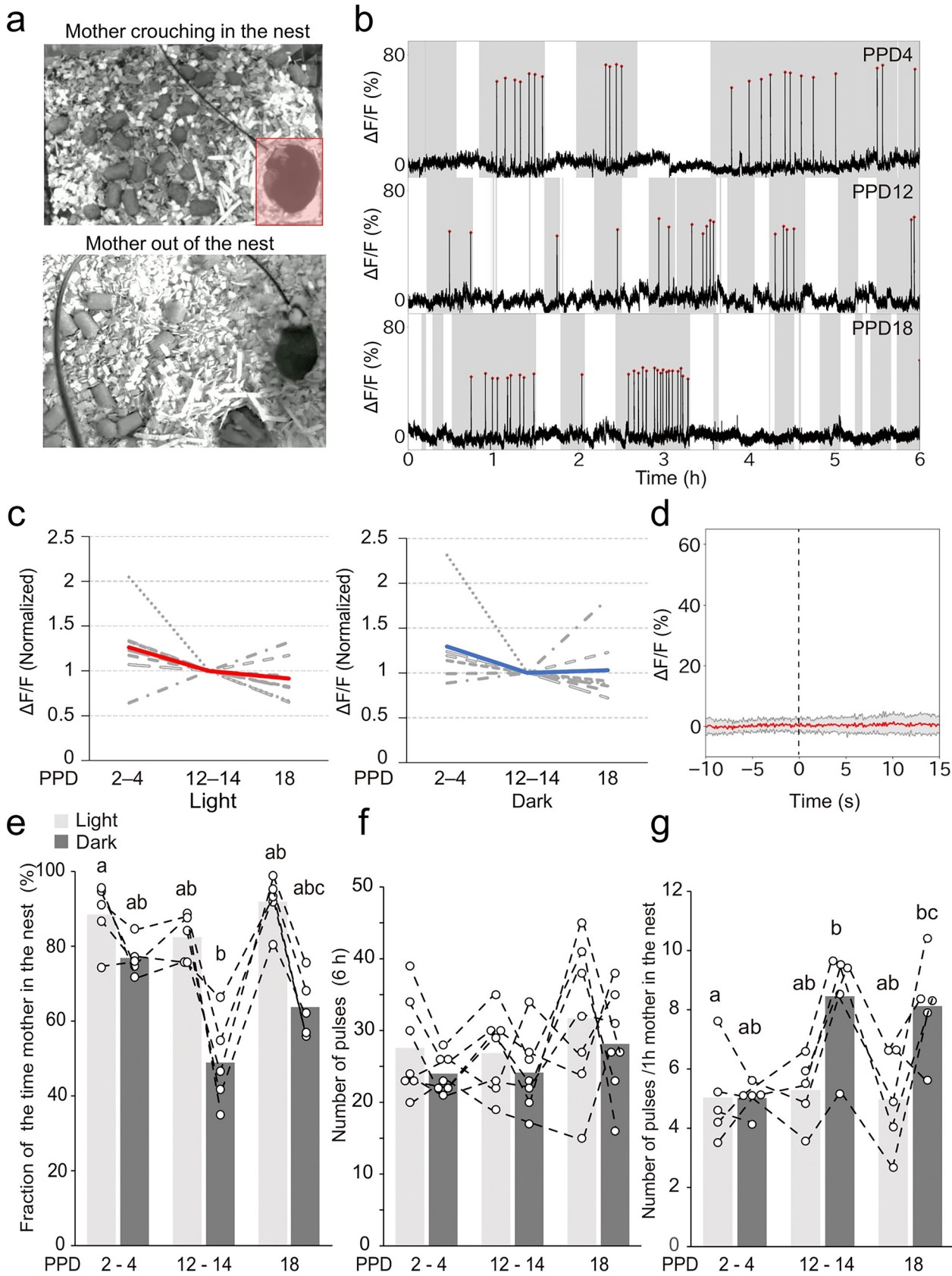

**Fig 4. Dynamics of mother–pup interactions and the pulsatile activities of OT neurons.** a. Examples of the mother mouse crouching over the pups in the nest (top) and being outside the nest (bottom). The red area shows the nest. b. Typical examples of photometry traces from the early (PPD2), middle (PPD13), and late (PPD18) lactation stages. Red dots represent the pulsatile activities of OT neurons. Data were recorded for 6 h during the dark phase. Shadows represent the time when the mother was in the nest while white areas represent the time when the mother was out of the nest. c. Intensities of pulsatile activity within individuals by normalizing the ΔF/F values to the data of PPD12–14 during the light (left) and dark (right) phases. The gray dotted lines indicate individual values, while the red and blue lines indicate average values. No statistical significance was found by two-sided one-way ANOVA. n = 7 mice. d. Peri-event photometric traces of PVH OT neurons during pup retrieval, where time zero was defined as the onset of pup retrieval. The red line shows the mean of 41 retrieval behaviors obtained from n = 3 mother mice, with the shadow representing the standard deviation. e. Quantification of the fraction of time the mother spent in the nest. n = 5 mice. f. Quantification of the number of photometric peaks per 6 h of recording. n = 7 mice. g. Quantification of the number of photometric peaks per 1 h of the mother in the nest. n = 5 mice. Different letters (a, b, and c) in panels e–g denote a significant difference (p < 0.05) by two-sided two-way ANOVA followed by a post hoc Tukey's HSD test.

that the pulsatile activities of OT neurons within the cluster follow a Poisson process, occurring randomly and independently from other pulsatile activities. Collectively, these data revealed temporal dynamics of pulsatile activities of OT neurons during the progression of natural lactation processes of wild-type mice.

## The age of pups affects the pulse distribution

Our data have revealed a difference in IPI distributions between the early and late lactation stages, which could be attributed to autonomous changes in the mother's neuroendocrine system or the behavior of pups. To assess the influence of the age of pups on IPI patterns, we presented foster pups of different ages to lactating mothers. First, elder pups (PND17–19) were introduced to the mothers during the early lactation stage (PPD2–4). Second, younger pups (PND2–4) were presented to mothers during the late lactation stage (PPD17–19). We monitored photometry traces for 6 hours during the light phase over three consecutive days: Day 1 with the original pups and Days 2 and 3 with the foster pups (Fig 6A). We analyzed the distributions of IPIs, the number of pulsatile activities per 6-hour window, and the magnitude of ΔF/F. When the elder pups were introduced to the mothers at PPD3 or 4, the proportion of short IPIs considerably increased (Fig 6B, top), mimicking the pattern observed during the late lactation stage. Conversely, when the young pups were introduced to the mothers at PPD18 or 19, the fraction of short IPIs moderately decreased, and longer IPIs (> 3000 s) vanished (Fig 6B, bottom), resembling the pattern seen in the early lactation stage. No significant differences were found in the number of pulsatile activities or the magnitude of ΔF/F values (Fig 6C and 6D). Collectively, these data suggest that the clustering of pulsatile activities during the late lactation stage is likely provoked, at least in part, by elder pups, although we do not exclude the possibility of autonomous changes in the mother's brain.

## Dynamics of pulsatile activities of OT neurons during peri-weaning stages

To characterize the dynamics of pulsatile activities of OT neurons during peri-weaning stages toward the termination of lactation, we next conducted daily fiber photometry recording of OT neurons for 6 h each in the light and dark phases on PPD18–27 (Fig 7A). The photometric peaks were clustered on PPD18, but became sparser around PPD22 and disappeared around PPD26. Quantitative analysis revealed that the number of peaks per 6 h period significantly decreased after PPD23, and the photometric peaks completely disappeared by PPD26 (Fig 7B). Although this trend was commonly observed between the light and dark phases, the reduction of peak numbers was significantly greater in the dark phase (Fig 7B and 7C). As the intensities of photometry peaks assessed by the maximum ΔF/F values differed among individuals, we next analyzed the maximum ΔF/F for each data point normalized to the mean maximum ΔF/F of the same animal on PPD18 (Fig 7D). The intensities were grossly constant throughout the

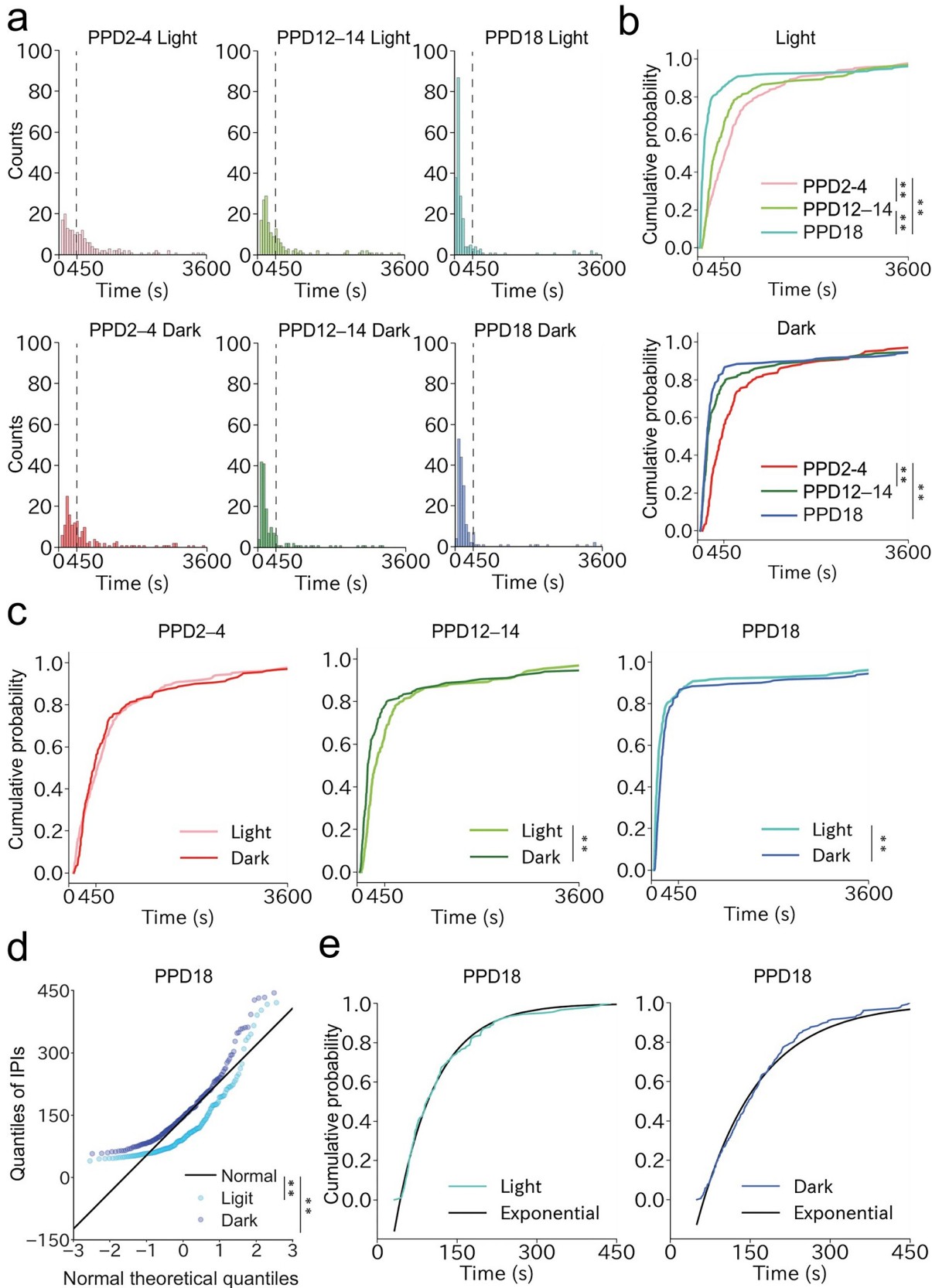

**Fig 5. Quantitative analysis of the dynamics of photometric peaks among different lactation stages.** a. Histograms of IPIs at PPD2–4, PPD12–14, and PPD18 in the light and dark phases. One bin is 60 s. b. Cumulative distribution of IPIs in the light (top) and dark (bottom) phases to compare patterns observed in PPD2–4, PPD12–14, and PPD18 mothers. **, p < 0.01 by the Kolmogorov–Smirnov test with Bonferroni correction. c. Cumulative distribution of IPIs in PPD2–4 (left), PPD12–14 (middle), and PPD18 (right) mothers to compare patterns observed in the light and dark phases. **, p < 0.01 by the Kolmogorov–Smirnov test. d. A quantile-quantile plot to visualize normality in the IPI distribution within 450 s on PPD18. **, p < 0.01 by Shapiro-Wilk Test. e. Comparison of the observed cumulative distributions of IPIs within 450 s in the light (left) and dark (right) phases on PPD18 with that of a computer simulation based on exponential distributions fitted by the least-squares method. n = 7 mice.

peri-weaning stage. Similarly, the waveform of pulses was also grossly unchanged (Fig 7E). Taken together, these data demonstrate that, during the peri-weaning stages, the dark phase precedes the light phase in a gradual reduction of pulsatile activities of OT neurons.

Does the gradual reduction in the pulsatile activity of OT neurons during the peri-weaning stage reflect a gradual decrease in mother–juvenile interaction? To obtain a hint, we analyzed the video data collected at PPD18, 21, and 24. We found that the amount of time the mother spent in the nest remained mostly unchanged during these time points, with longer durations in the light phase than in the dark phase (Fig 7F). As the number of pulses per 6-hour time window gradually decreased (Fig 7B), the number of pulses per one hour that the mother spent in the nest also decreased significantly as the weaning process progressed, with a trend towards higher pulse frequency during the dark phase (Fig 7G). These data suggest that, under our experimental conditions, weaning occurs without a drastic reduction in the mother's presence in the nest. Nevertheless, a more detailed analysis of direct mother–juvenile interaction will be required in the future to fully understand the mechanisms of weaning (see Discussion).

## Discussion

Lactation is a complex biological process involving bidirectional interactions between the maternal neuroendocrinological system and mother–infant behaviors, abnormalities in which can lead to mortality in offspring, particularly in low-resource settings. Despite the clinical importance and direct involvement in the quality of life of postpartum women, basic neuroscience studies of lactation processes remain limited. The present study expands the utility of chronic $Ca^{2+}$ imaging to monitor the maternal pulsatile activities of OT neurons in free-moving animals by a simple Cre-free AAV driven by an *OT* mini-promoter. Here, we discuss new biological insights obtained from this study and its future applications.

First, our data establish that strict physical contact with nipples by pups is required for the milk ejection reflex in mice. Even in well-experienced PPD12–14 mothers, no pulsatile activities of OT neurons occurred during physical separation (Fig 3) or when the mothers were outside the nest during spontaneous lactation (Fig 4). There is likely a species difference because lactating women and livestock animals can learn infant-related signals to eject milk before physical suckling [9, 10]. Although there is still room to force associative learning by operant conditioning paradigms in mice, our data suggest that under spontaneous lactation, mother mice do not form conditional learning of milk ejection to pup-associated sensory signals.

Second, by combining fiber photometry recordings with video-recording analysis of the locomotion of mothers, the temporal dynamics of pulsatile activities of OT neurons and mother–pup interactions were revealed throughout the lactation stages. Regarding the magnitude of pulsatile activity of OT neurons, although our previous study demonstrated that the peak height increased from PPD1 to PPD12–14 [14], no major change was observed between PPD2–4 and PPD12–14 in the current study. This suggests a substantial increase in peak height during the first one or two days of lactation. Regarding the temporal dynamics of pulsatile activities of OT neurons, in the early stage of lactation, mothers spend comparable time in

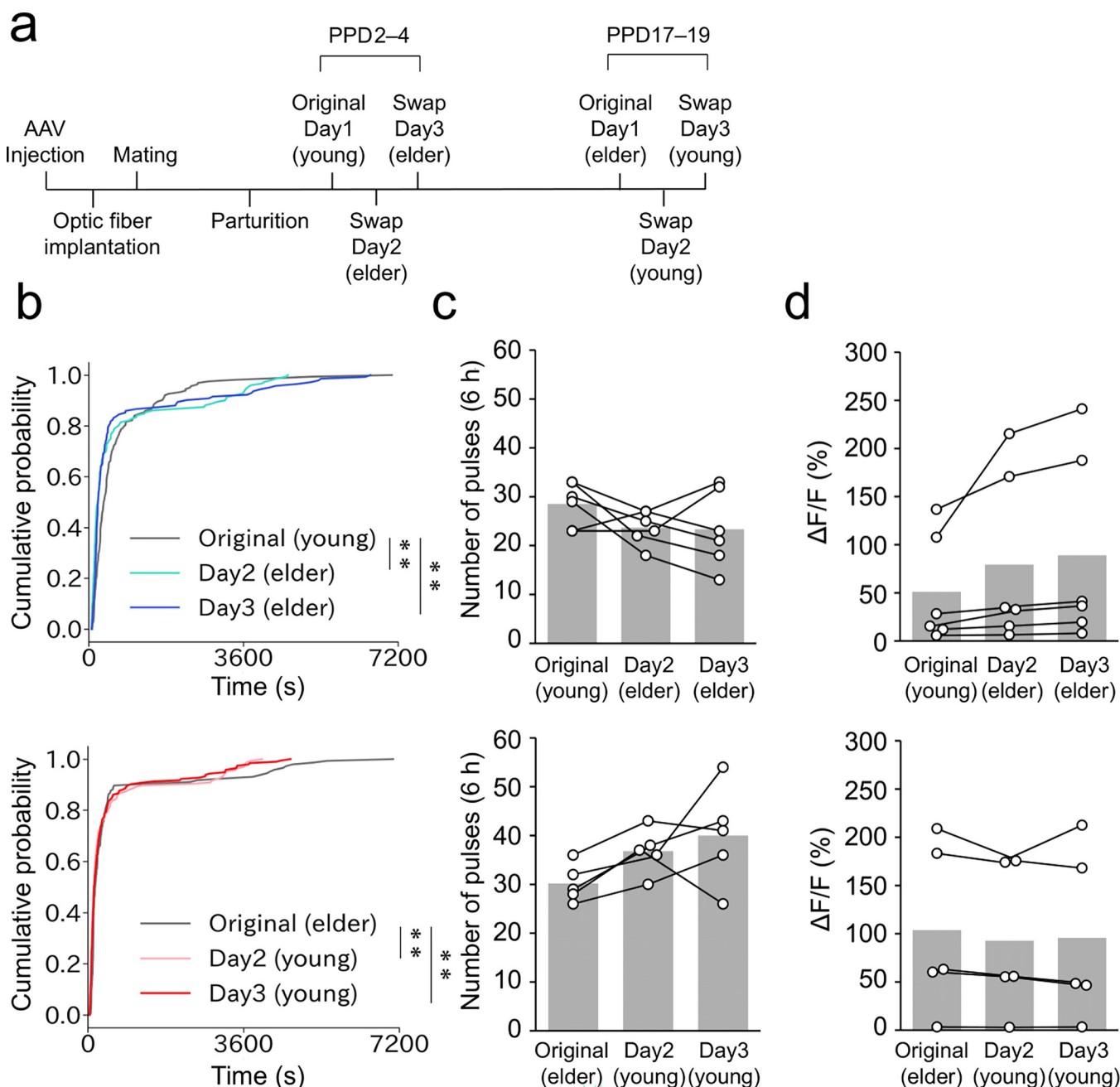

**Fig 6. Age of pups affects the pulse distribution.** a. Schematic of the timeline of the experiment. We defined young pups as PND 2–4 and elder pups as PND17–19. b. Cumulative distribution of IPIs in PPD2–4 (top) and PPD17–19 (bottom) mothers to compare the IPIs distribution with the original pups (Day 1) and foster pups (Day 2 and Day 3). **, p < 0.01 by the Kolmogorov–Smirnov test with Bonferroni correction. c. Quantification of the number of photometric peaks per 6 h of recording in PPD2–4 (top) and PPD17–19 (bottom) mothers. No significant difference was observed by two-sided one-way ANOVA. d. Quantification of the magnitude of ΔF/F in PPD2–4 (top) and PPD17–19 (bottom) mothers. No significant difference was observed by two-sided one-way ANOVA. n = 6 in PPD2–4 (top) and n = 5 in PPD17–19.

the nest for breastfeeding, regardless of the light or dark phase. Breastfeeding patterns are grossly constant in the light phase. In contrast in the dark phase, as lactation precedes, mothers spend less time in the nest, but provide more intensive breastfeeding (Fig 4), resulting in more clustered patterns of pulsatile activities of OT neurons (Fig 5). Our quantitative analyses

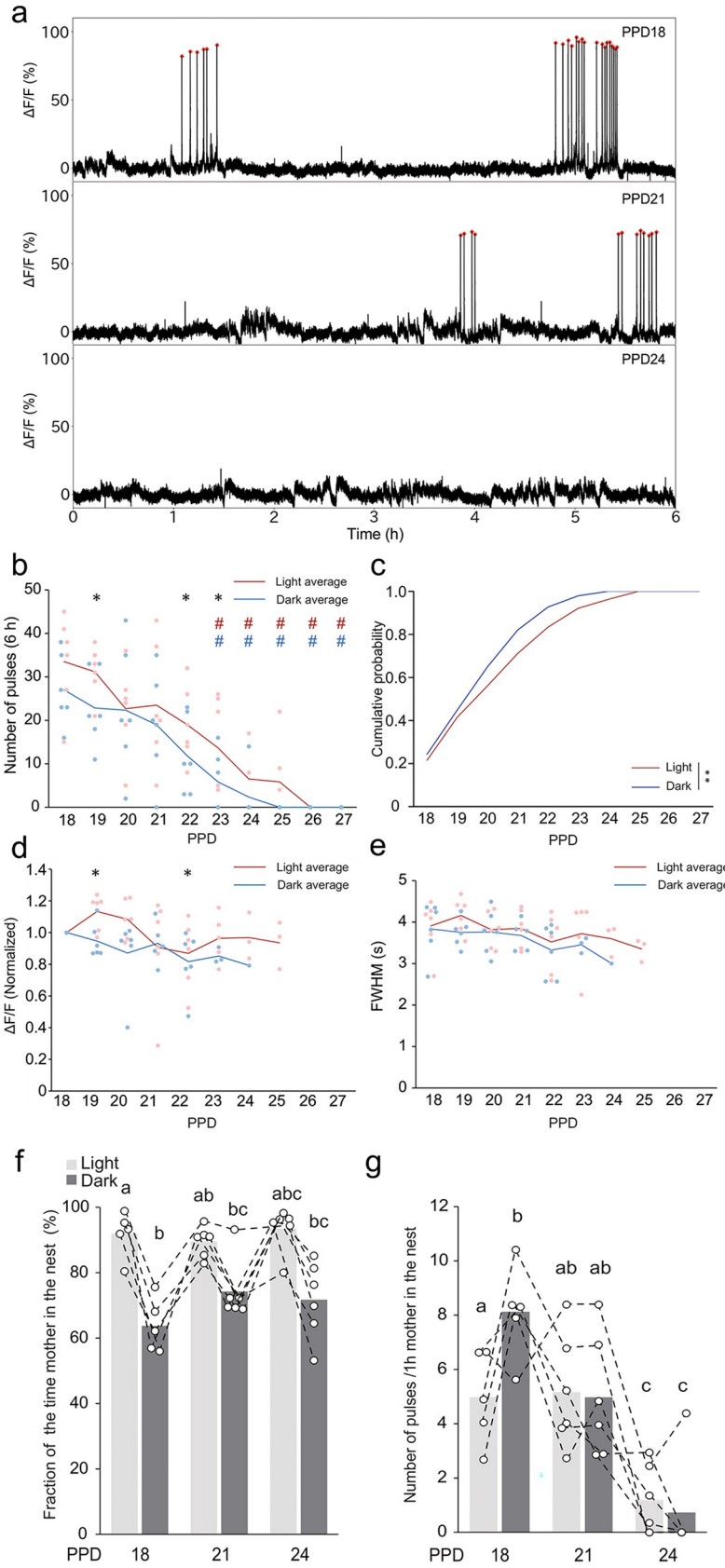

**Fig 7. Dynamics of the pulsatile activities of OT neurons during peri-weaning stages.** a. Typical examples of the photometry traces on PPD18, PPD22, and PPD26. Red dots indicate the pulsatile activities of OT neurons. b. Quantification of the number of photometric peaks per 6 h during the peri-weaning stages. Red and blue lines indicate mean values for the light and dark phases; red and blue dots indicate corresponding individual data. *, $p < 0.05$ by Wilcoxon signed-rank sum test, showing a statistical difference between the light and dark phases. Red # and blue # indicate statistical differences from PPD18 for the light and dark phases, respectively, by two-sided two-way ANOVA followed by a post hoc Tukey's HSD test. n = 6 mice. c. Cumulative distribution of the photometric peaks in light (red line) and dark (blue line) phases during the peri-weaning stages. **, $p < 0.01$ by the Kolmogorov–Smirnov test. d. Quantification of intensities of the photometric peaks in the weaning stages assessed by the normalized peak height to the mean peak height on PPD18. Symbols and statistics are the same as in panel b. e. Quantification of waveforms by FWHM during the same weaning stages. Symbols and statistics are the same as in panel b. f. Quantification of the fraction of time the mother spent in the nest. n = 5 mice in PPD18 and n = 6 mice in PPD21and 24. g. Quantification of the number of photometric peaks per 1 h of the mother in the nest. n = 5 mice in PPD18 and n = 6 mice in PPD21and 24. PPD18 data in panels f and g are the same as those shown in Fig 4. Symbols and statistics in panels f and g are the same as in Fig 4E–4G.

suggested that the occurrence of photometric peaks within the cluster followed a Poisson process, although underlying molecular and/or neural circuit mechanisms should await future studies. Based on fostering experiments, we further provided evidence supporting that the clustering of pulsatile activities during the late lactation stage is likely provoked, at least in part, by elder pups (Fig 6). One interpretation of this observation is as follows. In the early stage of lactation, neonatal infants constantly require milk, resulting in similar breastfeeding patterns between the light and dark phases. As pups grow and their single meal size increases, nocturnal mother mice can spend more time in non-lactation activities in the dark phase, such as feeding or investigations, resulting in a shorter and concentrated lactation period.

Third, during the peri-weaning stages (PPD22–26), the number of pulsatile activities of OT neurons decreases gradually, rather than suddenly, without changing the intensities or waveforms (Fig 6). The dark phase precedes the light phase in the reduction of pulsatile activities, suggesting that weaning precedes in the dark phase. This may be an example of parent–offspring conflict [28]: when the mothers are active in the dark phase, they can avoid nonessential breastfeeding; however, when they sleep in the light phase, pups of weaning age can "steal" their milk. However, our video data analysis did not support the idea that mothers actively avoid staying in the nest during the peri-weaning period (Fig 7F and 7G). Nevertheless, it is important to note that our video analysis only provides information regarding the mother's presence in the nest (mostly touching with pups), but not the precise duration of suckling by the pups. To capture the latter, a high-magnification video taken from the bottom of the cage is required, which presents a significant challenge, even during the earlier lactation stages when the pups are relatively immobile. This approach has only been successfully employed for brief durations as reported in [18]. Hence, our assessment of the time the mother spends in the nest (Figs 4 and 7) can only serve as a surrogate measure of mother–pup interaction. To obtain a better understanding of the mother–juvenile interaction during the peri-weaning period, future studies incorporating automated video analysis will be required.

Besides the ecological interpretation, it is also important to ask about the neurological mechanisms of weaning. Does the sensitivity of OT neurons to a given amount of nipple stimulation change during the weaning stage? How do the exact timings after the parturition and the mother's cognition of the age of pups affect the activities of OT neurons? Answering these questions could provide new insights into the mechanism of temporal specificity of the intensive milk ejection reflex observed during only the lactation periods.

Lastly, our previous study [22] and the present study have demonstrated the ability of the mouse 2.6 kb *OTp* to selectively target PVH OT neurons in C57BL/6 mice of various conditions, such as virgin males, virgin females, lactating mothers, and fathers. Furthermore, previous research has confirmed that a similar *OTp* can selectively label OT neurons in rats [21, 25].

Therefore, it is reasonable to consider that our AAV *OTp-GCaMP6s*-based imaging method of OT neurons is broadly applicable to various mouse strains, including ICR mice (Fig 2), natural mutant mice, and genetically modified mice, unlike our previous strategy, which relied on the complex genetic regulation of GCaMP6s [18]. As a proof-of-concept, without genetic crossing, we compared the temporal structure of pulsatile activities of OT neurons in two mouse strains in the early lactation stage (Fig 2).

Future studies should examine *OT* or *OTR* conditional knockout mice [3, 6] to pinpoint the functions of the OT-mediated facilitation of the milk ejection reflex [11, 12]. A mutant mouse line with abnormal milk compositions or deficits in infant-directed bonding might be of interest as a model of difficulties in maternal care. As our strategy is compatible with Cre-based cell type-specific neural manipulations such as opto- and pharmaco-genetics [29], we expect to facilitate further studies of afferent circuitry from the nipple to the OT neurons in terms of cell-type resolution. Furthermore, in principle, our strategy is applicable to many other mammalian species beyond mice and rats [8] for characterizing the neural activities of OT neurons during lactation. Such comparative studies could be expected to be beneficial in terms of gaining a better understanding of the common schemes and species specificity of lactation, which is a fundamental characteristic of all mammals, including humans.

## Acknowledgments

We thank the RIKEN BDR animal facility staff for caring for the animals, Yohsuke Fukai for the discussion on statistical and quantitative analyses, and the members of the Miyamichi Laboratory for their technical support and critical reading of the manuscript.

## Author Contributions

**Conceptualization:** Hiroko Yukinaga, Kazunari Miyamichi.

**Data curation:** Hiroko Yukinaga, Kazunari Miyamichi.

**Funding acquisition:** Kazunari Miyamichi.

**Investigation:** Kasane Yaguchi, Mitsue Hagihara, Kazunari Miyamichi.

**Methodology:** Kasane Yaguchi, Mitsue Hagihara, Ayumu Konno, Hirokazu Hirai.

**Writing – original draft:** Kasane Yaguchi, Kazunari Miyamichi.

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
