## [Decision Letter · Decision Letter 0]

16 Nov 2022

PONE-D-22-29163Dynamic Modulation of Pulsatile Activities of Oxytocin Neurons in Lactating Wild-type MicePLOS ONE

Dear Dr. Miyamichi,

Thank you for submitting your manuscript to PLOS ONE. After careful consideration, we feel that your manuscript will likely be suitable for publication if it is revised to address the points raised during the review process.

I am enclosing the comments that two reviewers made on your paper. As you will see, the overall assessment of both reviewers is positive. However, there are a number of issues that should be addressed before the paper can be accepted for publication. In particular, Reviewer#1 have raised concerns regarding the statistical analysis of the dataset. Please clarify this important point, and address other insightful comments from both reviewers.

We look forward to receiving your revised manuscript.

Kind regards,

Julieta Alfonso, Ph.D.

Academic Editor

PLOS ONE

“We thank the RIKEN BDR animal facility staff for caring for the animals, Yohsuke Fukai and the members of the Miyamichi Laboratory for their technical support and critical reading of the manuscript. This study was supported by the program for Brain Mapping by Integrated Neurotechnologies for Disease Studies (Brain/MINDS, JP21dm027111) from Japan Agency for Medical Research and Development (AMED) to H.H. and by KAKENHI (20K20589 and 21H02587) from the Japan Society for the Promotion of Science (JSPS) to K.M.”

“This study was supported by the program for Brain Mapping by Integrated Neurotechnologies for Disease Studies (Brain/MINDS, JP21dm027111) from Japan Agency for Medical Research and Development (AMED, https://brainminds.jp/en/) to H.H. and by KAKENHI (20K20589 and 21H02587) from the Japan Society for the Promotion of Science (JSPS, https://www.jsps.go.jp/english/e-grants/index.html) to K.M. The funders had no role in study design, data collection and analysis, decision to publish, or preparation of the manuscript.”

Reviewers' comments:

Reviewer's Responses to Questions

**Comments to the Author**

1. Is the manuscript technically sound, and do the data support the conclusions?

Reviewer #1: Yes

Reviewer #2: Yes

2. Has the statistical analysis been performed appropriately and rigorously? 

Reviewer #1: No

Reviewer #2: Yes

3. Have the authors made all data underlying the findings in their manuscript fully available?

Reviewer #1: Yes

Reviewer #2: Yes

4. Is the manuscript presented in an intelligible fashion and written in standard English?

Reviewer #1: Yes

Reviewer #2: Yes

5. Review Comments to the Author

Reviewer #1: In this interesting and compact study, Yaguchi et. al. used a newly generated AAV that express GcaMP6f in OT cells to record OT pulsatile activity during lactation. They demonstrated that OT pulsatile activity requires suckling in mice and revealed interesting dynamic changes in OT pulses over lactation days. The study is well performed and the tool will be useful for the neuroscience community. The manuscript is clearly written. There are only a few minor suggestions.

1. Figure 3C, error bar is missing. Please make sure that the data passes normality test before using t-test. This may not be the case given that the n is small, and the pulse number/magnitude of one animal is quite different from the rest three animals. For Number of pulses, it is not clear what the duration is. Please changes it to frequency of pulses or number of pulses/XX time.

2. Figure 4b, please indicate on the figure what gray and white shades mean.

3. In Figure 4, is there any change in pulse magnitude across lactation days? From the representative trace, it looks like the pulse magnitude is higher during early lactation. Is that true across animals?

4. Presumably, the same animal was recorded across days and light/dark cycle. All data should be paired. Please use lines to connect the dots that belong to the same animal. In such a way, it will be easier to see changes in behavior or pulse-related parameters across time. Two-way ANOVA (factor 1: light vs. dark; factor 2: PPD age) is more appropriate for analyzing Figure 4c-e.

5. As mentioned in the discussion, an interesting question is whether the change in IPI over lactation days is due to changes in OT system or changes in suckling behavior from the pups. As the pup age, presumably the suckling force will be different. This can be addressed by using older pups in early lactating females and younger pups in late lactating females. The authors are encouraged to address this question in the current study as it is quite related and straightforward.

6. The last figure was cut off at the bottom. Does the gradual decrease of OT pulses during peri-weaning stage reflect a gradual decrease of pup suckling? In other words, is the decrease in pulse frequency parallel to a decrease in female crouching time in the nest?

Reviewer #2: The authors described methods of photometric chronic measurements of calcium imaging by use of AAV vector that induces GCaMP6 expression under the control of a 2.6 kb mouse oxytocin mini-promoter. They found that pulsatile activities of oxytocin neurons require physical contact. The data are solid and very interesting.

Following points should be considered carefully.

1. Expression of GCaMP in oxytocin neurons was confirmed in non-pregnant females (C57BL/6N). This specific expression is also true in lactating animals and in ICR mice? At least discussion is necessary.

2. The authors showed that oxytocin pulses occurred during the period when mothers were in the nest (quantitative data are preferable, if there are), suggesting that direct contact may be necessary for oxytocin pulses. However, is there any direct evidence for necessity of direct stimulation of “nipple”? Pulses occurred only during suckling not during just contact with pups? Are there no pulsatile activities during maternal behaviors of retrieval or licking?

3. The authors showed that no oxytocin pulse was found under condition of indirect contact via mesh in the latter stage of lactation. Have you ever checked lactation mice that previously experienced lactation (mice of second or third lactation)?

6. PLOS authors have the option to publish the peer review history of their article (what does this mean?). If published, this will include your full peer review and any attached files.

Reviewer #1: **Yes: **Dayu Lin

Reviewer #2: No

---

## [Author Response · Author response to Decision Letter 0]

12 Apr 2023

Please see our Response to Reviewer document

---

## [Editor Report · Decision Letter 1]

27 Apr 2023

Dynamic Modulation of Pulsatile Activities of Oxytocin Neurons in Lactating Wild-type Mice

PONE-D-22-29163R1

Dear Dr. Miyamichi,

We’re pleased to inform you that your manuscript has been judged scientifically suitable for publication and will be formally accepted for publication once it meets all outstanding technical requirements.

Kind regards,

Julieta Alfonso, Ph.D.

Academic Editor

PLOS ONE

Additional Editor Comments (optional):

The authors have successfully addressed the reviewer's comments.
---

## [Editor Report · Acceptance letter]

2 May 2023

PONE-D-22-29163R1 

Dynamic Modulation of Pulsatile Activities of Oxytocin Neurons in Lactating Wild-type Mice 

Dear Dr. Miyamichi:

I'm pleased to inform you that your manuscript has been deemed suitable for publication in PLOS ONE. Congratulations! Your manuscript is now with our production department. 

Kind regards, 

on behalf of

Dr. Julieta Alfonso 

Academic Editor

PLOS ONE